# Electrolysis as a Universal Approach for Isolation of Diverse Chitin Scaffolds from Selected Marine Demosponges [note 1]

**DOI:** 10.3390/md20110665

**Published:** 2022-10-25

**Authors:** Krzysztof Nowacki, Maciej Galiński, Andriy Fursov, Alona Voronkina, Heike Meissner, Iaroslav Petrenko, Allison L. Stelling, Hermann Ehrlich

**Affiliations:** 1Institute of Chemistry and Technical Electrochemistry, Poznan University of Technology, Berdychowo 4, 60-965 Poznan, Poland; 2Institute of Electronics and Sensor Materials, TU Bergakademie Freiberg, Gustav-Zeuner Str. 3, 09599 Freiberg, Germany; 3Department of Pharmacy, National Pirogov Memorial Medical University, 21018 Vinnytsia, Ukraine; 4Department of Prosthetic Dentistry, Faculty of Medicine, University Hospital Carl Gustav Carus of Technische Universität Dresden, Fetscherstraße 74, 01307 Dresden, Germany; 5Department of Chemistry and Biochemistry, The University of Texas at Dallas, 800 W Campbell Rd, Richardson, TX 75080, USA; 6Center of Advanced Technology, Adam Mickiewicz University, Uniwersytetu Poznańskiego 10, 61-614 Poznan, Poland

**Keywords:** chitin, electrolysis, demineralization, marine sponges, *Aplysina archeri*, *Ianthella basta*, *Suberea clavata*

## Abstract

Three-dimensional chitinous scaffolds often used in regenerative medicine, tissue engineering, biomimetics and technology are mostly isolated from marine organisms, such as marine sponges (Porifera). In this work, we report the results of the electrochemical isolation of the ready to use chitinous matrices from three species of verongiid demosponges (*Aplysina archeri*, *Ianthella basta* and *Suberea clavata*) as a perfect example of possible morphological and chemical dimorphism in the case of the marine chitin sources. The electrolysis of concentrated Na_2_SO_4_ aqueous solution showed its superiority over the chemical chitin isolation method in terms of the treatment time reduction: only 5.5 h for *A. archeri*, 16.5 h for *I. basta* and 20 h for the *S. clavata* sample. Further investigation of the isolated scaffolds by digital microscopy and SEM showed that the electrolysis-supported isolation process obtains chitinous scaffolds with well-preserved spatial structure and it can be competitive to other alternative chitin isolation techniques that use external accelerating factors such as microwave irradiation or atmospheric plasma. Moreover, the infrared spectroscopy (ATR-FTIR) proved that with the applied electrochemical conditions, the transformation into chitosan does not take place.

## 1. Introduction

Chitin is a very ancient and abundant polysaccharide of natural origin [1,2,3,4]. This aminopolysaccharide can be found in an enormous number of diverse organisms such as yeast, fungi, protists, diatoms, sponges, corals, mollusks and annelids, and in the exoskeletons of crustaceans, insects and arachnids [5,6,7,8,9,10,11,12,13,14,15,16,17]. To date, the crustaceans’ shells obtained as a waste product from processing of Antarctic krill and other seafood are the main sources of chitin in the form of powders, flakes and fibers [3,8,18,19,20,21]. However, ready to use three dimensional (3D) chitinous scaffolds of diverse size, shape and porosity have mostly been isolated from marine sponges (Porifera) [22,23,24,25,26]. Recently, using chitin as a 3D scaffolding has been recognized as one of the intriguing strategies in modern bioinspired materials science [27,28].

Due to the broad variety of chitin applications in biomedicine [29] tissue engineering [30], technology [31,32,33,34] and biomimetics [17,35,36,37,38,39,40], this structural polysaccharide is being intensively studied, including the methodology of its isolation from diverse natural and mostly renewable sources [41].

The current technology of chitin isolation from crustaceans on an industrial scale is focused on the chemical extraction method which requires three time-consuming steps (demineralization, deproteinizaition and depigmentation) and generates a great excess of hazardous effluents [8,42]. Chitin-based biomineralization of invertebrates’ exoskeletons is a well-recognized phenomenon [43,44]. Consequently, the demineralization step is focused on the elimination of the water-insoluble calcium salts by the acidic treatment of the chitin precursor [45]. The deproteinization step uses concentrated alkaline solutions and increased temperature, which causes the hydrolysis of most of the proteins in the used chitin-based biological materials [6,46]. As the structure of the crustaceans’ shells is often characterized as a layered protein/mineral composite, the demineralization and deproteinization steps usually have to be repeated in order to achieve full isolation of the chitinous structure [6,46]. The depigmentation step is an optional treatment carried out by adding highly reactive oxidizing agents such as hydrogen peroxide. This step can be performed both to remove pigments from the chitinous skeleton and to hydrolyze residual proteins [6,46]. Unfortunately, the long treatment time followed by the substantial amount of chemicals that have to be used are main drawbacks of this classical chemical approach in chitin isolation. Therefore, increased attention of experts from industry and science are paid to develop new sustainable pathways to isolate naturally pre-structured chitinous scaffolds from marine sources. Based on the classical chemical extraction of chitin, alternative methods have been developed, and among them, the most promising approaches are ones that use external accelerating factors such as microwave irradiation or atmospheric plasma [47,48]. These assisted methods are focused on the use of additional treatment combined with the corresponding chemical method steps which results in easier and faster separation of proteins from chitinous skeleton [49]. Thanks to this phenomenon which is boosted by the synergic effect of the three-dimensional heating, the treatment time can be significantly reduced (from days to few hours) [47,48,50]. However, these methods seem to only have laboratory use, and their limitations in the case of large-scale process-like energy usage and overheating of the reaction mass cannot be omitted. Lately, another interesting approach based on the use of current flow as an external factor has been described [51]. The principle of this method was based on electrolysis of diluted NaCl aqueous solution to ensure acidic and alkali treatment of crustacean’s biomass. However, being based on the electrolysis of low concentrated NaCl solution, this approach was characterized by significant treatment time (13–19 h) [51]. Moreover, the evolution of chlorine gas on anode surface, which is a highly corrosive compound, was another drawback [51]. The modification of electrolysis-assisted isolation of chitin where the well-known chemical treatment was merged with in situ electrolysis of concentrated Na_2_SO_4_, was proposed in our previous research and gave great results in the case of reduction of time treatment and amounts of chemical used [52,53]. The principle of this method is electrolysis of concentrated aqueous electrolyte solutions to ensure acidic and alkali treatment for chitin precursors. Briefly, the water electrolysis process is a well-known electrochemical phenomenon that has to be thermodynamically forced by the flow of direct electric current from an external source [54,55]. In order to pass current between two electrodes, a specific electrolytic cell (electrolyzer) has to be constructed. The modern electrolyzer is composed from two symmetrical polarizable electrodes made from electro- and chemically inert materials. Usually both electrodes are dipped in an electrically conductive solution (electrolyte) and separated with an ion exchange membrane (cation, anion or bipolar) forming two compartments [56,57,58,59,60,61]. The chamber with the anode contains the electrolyte solution called an anolyte and the respective chamber with the cathode is filled with a catholyte. Aqueous solutions of low molecular salts such as Na_2_SO_4_, which occurs as by-products in numerous chemical processes, are the perfect substrates for production of alkali and acids by electrolysis. The splitting of Na_2_SO_4_ aqueous solution in a cation exchange membrane (CEM) electrolyzer into NaOH and H_2_SO_4_ solutions is one of the most popular ways to utilize an overproduction of this salt [62]. Figure 1A shows the basic principle of this process [63].

The fundamental electrochemical reactions that occur during Na_2_SO_4_ aqueous solution electrolysis (decomposition of water particles) have a place on the electrodes surface. On the anode surface:2H_2_O → O_2_ + 4H^+^ + 4e^−^(1)
and on the cathode surface:4H_2_O + 4e^−^ → 2H_2_ + 4OH^−^(2)

The result of this redox reaction is an excess of H^+^ ions in anolyte and respectively an excess of OH^−^ ions in catholyte. Simultaneously, the sodium ions from the anolyte that migrate through CEM go towards the cathode where they are reduced to sodium metal and immediately react with the water to form NaOH [64].

Due to this phenomenon, it is possible to establish and change the pH in each part of electrolyzer by simply applying a specific potential or change of the electrodes’ polarity. Thus, due to the real time pH control, the electrolysis-assisted isolation of chitin could be a very flexible process in terms of usage of different chitin sources [51,52,53]. Our preliminary experiments have shown that in contrast to traditional methods, the application of electrolysis will reduce the treatment time from 3–7 days approximately to 9–12 h [13,52,53,65].

In this study, we present for the first time the results of the electrochemical isolation of chitin from three different species of demosponges as a perfect example of possible morphological and chemical dimorphism in case of the marine chitin sources. The *Aplysina archeri* (cylindroid-formed marine demosponge), *Ianthella basta* (flat-formed marine demosponge) and *Suberea clavata* (with heavy mineralized fibroust chitin-based skeleton) were utilized as sources of unique forms of exoskeletal chitin in electrolysis-assisted isolation method. Since all three sponge species used in the experiment show different morphology and chemical composition, it was necessary to modify our approach in each case by changing the potential and treatment time. Moreover, in this study, a concentrated Na_2_SO_4_ aqueous solution was utilized as the electrolyte, and the electrochemical isolation method was investigated in terms of its usefulness for the extraction of chitinous skeletons from diverse marine sources in the future.

## 2. Results and Discussion

The selection of such marine demosponges of verongiida order such as *A. archeri*, *I. basta* and *S. clavata* as chitin sources was motivated by the fact that these sponge species have been relatively well described as the renewable sources of marine α-chitin in the chemical extraction method as well as in the other alternative isolation approaches [6,66,67,68,69]. Moreover, the differences in their morphology and chemical composition are a perfect example of the diversity of marine chitin sources, and give an opportunity to show the versatility of in situ electrolysis as a principal chitin isolation method [6,66,67,68,69,70,71,72,73].

Morphological changes during an electrochemically-assisted isolation process within the sponges were investigated with a digital optical microscope and scanning electron microscopy (SEM). In the first experiment with *A. archeri* as a chitin source, the isolation process was split into three main steps (respective to the chemical extraction method). Figure 2A displays the close-up of the examined sample before the first step of the electrochemical treatment. In this figure, the well-preserved structure of *A. archeri*, with clusters of cells supported by the chitin-based skeleton of tubular network architecture, can be observed. The first part of the treatment in the electrolytic cell was carried out in the cathode chamber for 1.5 h (12 V, 0.5 A) and the decellularization in the alkaline environment resulted in the complete dissolution of *A. archeri* somatic cells. Results of the first catholyte treatment are depicted in Figure 2B and revealed deep-yellow, semitransparent and cell-free *A. archeri* chitin-based scaffold. Next, in order to demineralize the sample, the anolyte treatment was applied. Decalcification was performed for 1.5 h (12 V, 0.5 A) and the microscopic images obtained after this step (Figure 2C) show that acidic environment allowed for removal of the carbonate salts and acid-soluble pigments within the sample. This anolyte treatment resulted in a light yellow, semitransparent *A. archeri* scaffold without significant deformations of the tubular network structure. The final part of the electrolytic treatment was applied to remove the remaining proteins, pigments and possible silica remnants, which were structurally incorporated into the chitinous tubes of the treated sample. Thus, the *A. archeri* chitin-based scaffold was placed in the cathode chamber for 2.5 h (16 V, 1.0 A). The microscopic investigation of post-treated sample revealed that the extremely high pH of the catholyte caused complete removal of pigments and residual proteins from the chitinous matrix, and Figure 2D presents a colorless *A. archeri* scaffold.

Further detailed morphology investigation of the *A. archeri* sample after full electrolytic treatment was performed by SEM. Figure 3A shows a well-preserved branched network of chitinous microtubes with only slightly harmed spatial structure without any somatic cells residuals. The local damages of the original bio-architecture of the chitin-based *A. archeri* skeleton was probably caused by intensive hydrogen evolution on the cathode. The surface of the single branch of the chitinous scaffold is presented in Figure 3B, where the characteristic grooved surface is clearly visible. The total time of *A. archeri* treatment in the electrolytic cell (5.5 h) was drastically reduced in comparison to the standard chemical extraction method (up to 7 days) and makes this isolation technique comparative even with the high energy-consuming methods such as microwave treatment [6,69].

Figure 4A,B display the detailed view of the examined *I. basta* sample before electrochemical treatment, where the homogeneous surface and characteristic spikes supported by the chitin-based skeleton can be observed. Due to the high content of calcified phase inside the *I. basta* chitinous skeletal tubes, the electrochemical approach to the isolation of the chitin scaffold had to be modified [67]. Decellularization was performed the same as *A. archeri* (1.5 h in cathode chamber, 12 V, 0.5 A) since the main goal of this step was only to decompose *I. basta* sponge somatic cells. The photographic image obtained after this part of the isolation process is presented in Figure 4C and revealed a dark brown, cell-free *I. basta* chitin-based skeleton in the form of a highly organized mesh of microtubes. Detailed investigation of the sample depicted in Figure 4D shows a close-up view of one of the branches with characteristic mineralized structures inside the *I. basta* fiber [67]. A further procedure of the electrochemical isolation was modified and performed as the repeatable steps, where one full treatment cycle consisted of the anolyte and catholyte treatment. Demineralization as well as deproteinization/depigmentation/desilicification were carried out using stable electrochemical conditions (12 V, 0.5 A) for 0.5 h each, and performed until a soft, colorless scaffold was obtained (15 cycles/15 h). A unique *I. basta* flat skeleton made of highly organized chitinous microfibers intercalated by calcitic nanocrystals has been especially resistant to the acid/base exposure [66,67]. However, the flexibility of the electrolysis-supported isolation method allowed rapid changes between acid/base treatment and ensured free access of the anolyte and catholyte to the layers soluble in the corresponding pH conditions. Gradual removal of the acid and base soluble chemical compounds from the *I. basta* skeletal fibers can be observed on the Figure 4E–H. The sample of *I. basta* after the first cycle of anolyte/catholyte treatment is depicted in Figure 4E and shows a light brown and still rigid, highly organized, branched skeleton. Further treatment caused gradual decolorization and partial loss of mechanical rigidity of the sample, which can be visible in Figure 4F (obtained after the eighth cycle of anolyte/catholyte treatment). Finally, after 16.5 h of summary treatment in the electrolytic cell the soft, colorless chitinous scaffold was obtained (Figure 4G,H). Further detailed morphology investigations of the isolated *I. basta* scaffold were performed by SEM technique. Figure 3C shows relatively good-preserved mesh of the chitinous microfibers with a slightly harmed spatial structure without any somatic cell residuals. However, the dissolution of the calcified phase within the *I. basta* skeleton and following loss of rigidity caused visible on the Figure 3C deformation of the original bio-architecture of the branched network of chitinous fibers. The surface of the single microtube is presented in Figure 3D, where the irregular grooved surface without any cracks or holes is clearly visible. Total time of *I. basta* treatment in the electrolytic cell (16.5 h) was significantly reduced in comparison to the standard chemical extraction method (up to 3.5 days) [66,67,70]. Moreover, this experiment shows that with proper modification of electrochemical approach it is possible to isolate chitin even from the marine sources with relatively high mineralization.

The approach used for *I. basta* was applied in order to isolate chitin from the *S. clavata* skeletal micro-tubes, since the previous literature reports described *S. clavata* skeletal fibers as heavy mineralized structures based on a chitinous template intercalated by calcium carbonate and poorly soluble silica [68]. The decellularization step in performed electrolysis-assisted isolation of chitin was omitted, due to the possibility of manual extraction of well-visible skeletal fibers from this sponge body. The dark yellow skeletal filament of *S. clavata* after separation from sponge somatic cells is presented in Figure 5A, where the branched architecture of the skeleton is clearly visible. In Figure 5B, the characteristic layered structure of the skeletal tube is clearly visible. Since the mineralized phase is mostly intercalated between the layers of the chitinous fiber, the decalcification and deproteinization/depigmentation/desilicification steps were repeated multiple times. Isolation process were performed by using stable electrochemical conditions (12 V, 0.5 A) and during one treatment cycle, the sample was placed for 1 h in both anode and cathode chambers. During the electrolytic treatment consisting of total 10 cycles, the successive changes in the acidic and basic environment led to the gradual depigmentation of the sample and loss of the mechanical rigidity, which can be observed on photographic images presented in Figure 5C–F. Finally, the colorless, soft tubes were obtained (Figure 5G,H) and subjected to the further morphology investigation by SEM. A close-up view to the *S. clavata* skeletal fiber morphology is displayed in Figure 3E, although the chitinous tube collapsed during drying, the surface remains unharmed without cracks or holes and shows an interesting tree bark texture. Investigation in higher magnification (Figure 3F) revealed that the entire surface of the sample was covered by tangled-up filaments. Total time of *S. clavata* treatment in the electrolytic cell (20 h) was also significantly reduced in comparison to the standard chemical extraction method (up to 4 days) [68]. Moreover, this experiment shows that with proper modification of the electrochemical approach, it is possible to dissolve the silica embedded in the chitinous matrix of *S. clavata* and eliminate the hazardous HF treatment from the procedure [68].

Calcofluor white (CFW) staining is a broadly applied method of determining the chitinous nature of the materials isolated from marine organisms [14,52,53,70,74]. Essentially, this technique relies on binding of the CFW molecules with the polysaccharides containing β-glycosidic bonds (such as chitin), and observation of this fluorochrome material under UV excitation where it should emit bright blue light even with a very short exposure time. Here, we used it as a fast and facile preliminary identification technique that provides us with valuable information about the presence of polysaccharides within *A. archeri*, *I. basta* and *S. clavata* samples. The visibly-enhanced blue light fluorescence after CFW staining showed in the Figure 6B,D,F clearly indicates that the material isolated in the electrolysis-supported method has a polysaccharide origin. Light exposure times were equal: 1/4800 s, 1/1900 s and 1/600 s for *A. archeri*, *I. basta* and *S. clavata*, respectively. These results are similar to the previously described CFW-based *A. archeri*, *I. basta* and *S. clavata* chitin identification [68,69,70].

Unfortunately, the CFW staining is an inconclusive method for chitin identification; therefore, the highly sensitive and well-established method for chitin investigation of ATR-FTIR spectroscopy was applied [75,76,77]. ATR-FTIR spectroscopy can confirm chitin, identify the crystalline form and distinguish the polysaccharide from chitosan [69,75,76]. ATR-FTIR spectra of *A. archeri* (green line), *I. basta* (red line) and *S. clavata* (purple line) scaffolds after the electrochemically-assisted isolation process are depicted in Figure 7. To exclude the possibility of deacetylation during the described process, the α-chitin standard (Figure 7, black line) was used as reference. All analyzed spectra of the investigated sponge scaffolds show as bands characteristic for α-chitin such as amide I (carbonyl stretching vibrations of N-acetyl groups), amide II (νN-H and νC-N) and amide III (νC–N and δN–H)–1612 cm^−1^, 1539 cm^−1^, 1302 cm^−1^, respectively, on the reference spectrum (black line). Since the location of all these characteristics for chitin bands on the spectra of investigated samples are nearly identical to α-chitin standard, it can be assumed that the transformation into chitosan did not take place. The distinctive features are the amide II and amide III bands which, in case of a strong deacetylation process, should be shifted to the approximate wavelength of 1587 cm^−1^ and 1319 cm^−1^, respectively [69,75,76]. While on the *A. archeri* (green line), *I. basta* (red line) and *S. clavata* (purple line) spectra, the bands are located much closer to the chitin reference: *A. archeri*; A^II^—1556 cm^−1^ and A^III^—1307 cm^−1^, *I. basta*; A^II^—1549 cm^−1^ and A^III^—1305 cm^−1^ and *S. clavata*; A^II^—1538 cm^−1^ and A^III^—1305 cm^−1^. Moreover, further detailed analysis of the isolated chitin scaffold spectra revealed that the characteristic band around 897 cm^–1^ (CH deformation of the β-glycosidic bond as well as C-O-C bridge) is still present, which suggests the occurrence of α-chitin in all samples (890 cm^−1^ for β-chitin) [7,78,79].

Results obtained using ATR-FTIR were further confirmed by the X-ray measurements. The diffraction patterns (Figure 8) revealed typical spacings for the α-chitin described in the literature, with two well-distinguished narrow peaks (at 2θ~9.0° and 2θ~19.2°) which can be interpreted as an ideal crystalline phase [17,36]. The detailed analysis of the X-ray diffractograms of all isolated chitin scaffolds showed that for *I. basta* and *S. clavata*, the characteristic diffraction peaks became broader and weaker, which indicate that a long-time electrochemical treatment causes degradation in the chitinous crystalline structure.

## 3. Conclusions

In this study, the in situ electrolysis of concentrated Na_2_SO_4_ solution was applied for the first time as an alternative isolation method of chitinous scaffolds and fibers from three morphologically different species of verongiid sponges: *A. archeri*, *I. basta* and *S. clavata*. The diversity of chitin sources used forced major changes in the procedure of the electrolysis-supported isolation process; however, the treatment times in all three cases were significantly shorter than the respective times in the standard chemical extraction method. The approach used in isolation of chitin from heavy mineralized *I. basta* and *S. clavata* sponges that involved cyclic changes of acidic and basic treatment gave astonishing results and the soft, colorless scaffolds/fibers were obtained. Moreover, the digital light microscopic investigation of isolated chitinous specimens revealed that despite minor mechanical damages, the final products of the electrochemical isolation process showed well-preserved spatial architecture with a characteristic branched microtubular network. Further investigation of the isolated scaffolds/fibers with Calcofluor white staining proved without a doubt that these materials were of polysaccharide nature and attenuated total reflectance. Fourier transform infrared spectroscopy confirmed that in all three cases, the pure α-chitin was obtained, and under applied conditions, the transformation of chitin to chitosan does not occur. All these features show that the method proposed in this study (in all variants) is an effective and time-saving alternative to the standard chemical extraction process and other novel high-energy consuming approaches (i.e., microwave irradiation treatment), even in the case of chitin isolation from the heavy mineralized marine sources. Previously, we have shown with strong evidence the absence of any kind of cytotoxic features of diverse chitinous scaffolds isolated from verongiid demosponges against such cells as human chondrocyte [80,81], human adipose tissue-derived MSCs [70,74,82] as well as human induced pluripotent stem cell-derived cardiomyocytes [30]. Therefore, further development of the electrolysis assisted isolation method as well as corresponding apparatuses for large-scale applications should be performed in the near future.

## 4. Materials and Methods

### 4.1. Biological Samples and Chemicals

Samples of dried marine demosponges *Aplysina archeria* and *Ianthella basta* (Figure 9A,B) were purchased from INTIB GmbH (Freiberg, Germany). The specimen of *Suberea clavata* Pulitzer-Finali, 1982, (Figure 9C,D) originate from Bali, Indonesia. It was collected on 19 April 2001 during the expedition on Bali Lombok Strait. Geographical coordinates of the locality: N side of Nusa Nembongan, Tanjung Taal (=Tanjung Ental = “Blue”), lat: 08°3933 S long: 115°2637 E; Field#: BAL.30/190401/237. The specimen was originally stored in ethanol in Naturalis Biodiversity Center, Leiden, the Netherlands, under Nr.RMNH.POR.1578.2.2. Sodium sulfate (Na_2_SO_4_, ≥99.7%) purchased from VWR (Darmsadt, Germany) was used for the preparation of aqueous electrolyte solution. Distilled water was used to prepare all aqueous solutions.

### 4.2. Electrolysis Cell Setup

The schematic illustration of the experimental system for the electrolysis-supported isolation of chitin is shown in Figure 1B. The CEM (cation exchange membrane) electrolyzer consisted of two cylindrical poly(propylene) chambers (50 mL each) separated by a PET reinforced cation exchange membrane Fumasep^®^ FKS-PET-130 (FUMATECH BWT GmbH, Bietigheim-Bissingen, Germany). Both electrodes were made of platinum wire (ø 0.5 mm and 5 cm in length) obtained from Carl Roth GmbH & Co. KG (Karlsruhe, Germany). The distance between them was kept at about 10 cm and they were connected to a DC power supply VoltCraft PS2043D (Conrad Electronic International GmbH & Co., Wels, Austria). 1.9 M sodium sulfate aqueous solution with an initial temperature of 40 °C was utilized as an anolyte as well as a catholyte and a digital pH meter Dostman 5040-0301 (Dostman electronic GmbH, Wertheim, Germany) was used for pH control of the electrolyte.

### 4.3. Isolation of Chitin Structures

The isolation of 3D chitinous scaffolds from *A*. *archeri* as well as *I. basta* and chitin-based skeletal fibers from *S. clavata* was realized by the electrolysis-supported isolation method. Differences in morphology and chemical composition of the used specimens forced the application of three separate procedures where the potential and treatment time were changed (Figure 10). It should be noted that the initial concentration of Na_2_SO_4_ for every procedure was 1.9 mol L^−1^ and the starting temperature for both anolyte and catholyte solutions was 40 °C.

#### 4.3.1. Isolation of Chitinous Scaffold from *A. archeri* Demosponge

The *A. archeri* sample (Figure 9A) in the pretreatment was cut into 0.5 g pieces and rinsed repeatedly with distilled water in order to get rid of major solid impurities and water-soluble salts of marine origin.

Step 1—Decellularization was carried out in the cathode chamber for 1.5 h (12 V, 0.5 A) (Figure 10A) and during this the electro-alkali treatment of the pH of catholyte solution was established up to 12.0. Hence, the rapid dissolution of preswelled cells followed by the removal of soft tissues from the interlayer spaces and a partial depigmentation of the sponge skeleton were observed. The sample after treatment was composed of a gold color cell-free skeleton with the net-like 3D structure (Figure 2B).Step 2—Decalcification was performed in the anode chamber for 1.5 h (12 V, 0.5 A); during this, the electro-acidic pH treatment of the anolyte solution was established down to 1.5. Low pH was necessary to purify the sponge skeleton from the mineral calcium salts and acid-soluble pigments. After treatment, the remaining sample was in the form of a light yellow skeleton without significant structure deformation (Figure 2C).Step 3—The deproteinization/depigmentation/desilicification processes were carried out after the exchange of the electrolyte in the cathode chamber due to high content of impurities that remained after dissolution of the cells in the decellurarization step. The sample was placed in the cathode chamber for 2.5 h (16 V, 1.0 A) and electro-alkali treatment the pH of catholyte solution was established up to 12.5. Lack of the possible barriers such as sponge cells or layers of the mineral salts gave the catholyte solution free access to the chitinous skeleton which, along with extremely high pH, caused incomplete removal of pigments and residual proteins from the chitinous matrix. After treatment, the remaining sample in form of a colorless scaffold (Figure 2D) was extensively rinsed using distilled water up to neutral pH and stored in ethanol absolute (4 °C).

#### 4.3.2. Isolation of Chitinous Scaffold from *I. basta* Demosponge

The *I. basta* sample (Figure 9B) in the pretreatment was cut into 0.2 g pieces and rinsed repeatedly with distilled water in order to get rid of major solid impurities and water soluble salts of marine origin.

Step 1—Decellularization of the *I. basta* was performed in the same way as in the *A. archeri* case (1.5 h in cathode chamber; 12 V, 0.5 A) and the post-treated sample was in the form of a rigid deep brown cell-free skeleton (Figure 4C,D).Step 2—Decalcification was carried out in the anode chamber for 0.5 h (12 V, 0.5 A) and during this, the electro-acidic treatment the pH of anolyte solution was established down to 1.5.Step 3—Deproteinization/depigmentation/desilicification was performed in the cathode chamber for 0.5 h (12 V, 0.5 A) and during this electro-alkali treatment of the pH of the catholyte solution was established up to 12.0.Step 2 and Step 3 cycle—Unique chemical composition of the *I. basta* demosponge skeleton [66,67] forced modification of the electrolysis-supported isolation method known from the earlier reports [52,53]. Due to the possible multilayered structure of proteins, pigments and biominerals within the *I. basta* skeleton, the decalcification and deproteinization/depigmentation/desilicification steps were repeated multiple times in order to ensure free access for the anolyte/catholyte to the layers soluble in corresponding pH conditions. Thus, a full 15 cycles consisting of the successive anolyte and catholyte treatment were performed (Figure 10B). During this process, the I. basta skeleton was gradually decolorized and softened until the colorless chitinous scaffold was obtained (Figure 4E–H). After treatment, the sample was extensively rinsed using distilled water up to neutral pH and stored in ethanol absolute (4 °C). Loss of the electrolyte caused by evaporation was compensated by refill of the anolyte and catholyte solutions every 5th cycle (5 h of treatment).

#### 4.3.3. Isolation of Chitinous Scaffold from *S. clavata* Demosponge

In pretreatment, the initial samples of *S. clavata* demosponge (Figure 9D) were cut out from the main sponge specimen (Figure 9C) and rinsed repeatedly with distilled water in order to eliminate major solid impurities and water soluble compounds of marine origin. Subsequently, the rigid, yellow colored skeletal fibers were manually isolated from sponge cells and once more rinsed with distilled water. Since the decellularization step was performed during sample preparation, it was possible to skip this part of the process in the electrolysis-supported isolation of the chitin procedure (Figure 10C). Hence, the electrolysis treatment of the *S. clavata* skeletal fibers started from the electro-acidic treatment in the anolyte solution.

Step 1—Decalcification was carried out in the anode chamber for 1.0 h (12 V, 0.5 A) and during this electro-acidic treatment the pH of anolyte solution was established down to 1.5.Step 2—Deproteinization/depigmentation/desilicification was performed in the cathode chamber for 1.0 h (12 V, 0.5 A) and during this electro-alkali treatment the pH of catholyte solution was established up to 12.0.Step 1 and Step 2 cycle—Due to similar reasons [68] in the *I. basta* case, the decalcification and deproteinization/depigmentation/desilicification steps were repeated multiple times. Hence, full 10 cycles consisting of the successive anolyte and catholyte treatment were performed (Figure 10C). During this process, the *S. clavata* skeletal fibers were gradually decolorized (Figure 5A–F) and had lost their mechanical rigidity till the colorless chitinous tubes were obtained (Figure 5G,H). Loss of the electrolyte caused by evaporation was compensated by the refill of the anolyte and catholyte solutions every 3 cycles (6 h of treatment). Obtained samples in the posttreatment were rinsed in distilled water up to a neutral pH and stored in ethanol absolute (4 °C).

### 4.4. Light and Fluorescence Microscopy 

*A. archeri*, *I. basta* and *S. clavata* demosponges samples at different stages of the chitin isolation process were investigated using a Keyence VHX-7000 digital optical microscope with zoom lenses VHX E20 (magnification up to 100×) and VHX E100 (magnification up to 500×) (Keyence, Osaka, Japan). Fluorescence microscopy images were obtained using Keyence BZ-9000 digital optical microscope (Keyence, Osaka, Japan).

### 4.5. Calcofluor White Staining

Calcofluor white (CFW, Fluorescent Brightener M2R, Sigma-Aldrich, St. Louis, MO, USA) staining was applied preliminarily to confirm the presence of chitin in materials isolated by the electrolysis-assisted method. This technique was used several times in different variants [9,14,70,74]. However, in this study, the 30 µL of a solution containing 10 g glycerin and 10 g NaOH in 90 mL of water was applied as a buffer. Subsequently, after 5 min, the CFW was added to the samples, and the investigated materials were incubated in staining mixture for 6 h without sunlight at 25 °C. After this treatment, samples were extracted from the staining solution and extensively rinsed with distilled water to eliminate the free unbounded molecules of CFW stain, then dried at 25 °C and investigated using fluorescent microscopy.

### 4.6. Attenuated Total Reflectance Fourier Transform Infrared Spectroscopy

The attenuated total reflectance Fourier transform infrared spectroscopy (ATR-FTIR) technique was used for the qualitative characterization of isolated chitinous scaffolds from *A. archeri* and *I. basta* and fibers from *S. clavata*. The presence of characteristics for α-chitin functional groups in obtained materials was analyzed using a Nicolet 210c spectrometer (Thermo Scientific, Waltham, MA, USA). All spectra were recorded using a wave number range of 4000–500 cm^–1^ (resolution of 1 cm^–1^). The samples before examination were dried to the dry mass in 40 °C.

### 4.7. Scanning Electron Microscopy

The surface morphology and microstructure of the chitinous scaffolds isolated from *A. archeri*, *I. basta* and *S. clavata* were analyzed using SEM images with a scanning electron microscope (XL 30 ESEM, Philips, Amsterdam, The Netherlands). Prior to scanning, the samples were coated with a gold layer using the Cressington Sputtercoater 108 auto, Crawley (GB) (sputtering time 45 s).

### 4.8. X-ray Diffraction 

The crystalline composition of samples were analyzed by X-ray diffraction using SEIFERT-FPM URD6 diffractometer equipped with a sealed X-ray tube with Cu anode and a secondary graphite monochromator placed in front of a proportional counter. The XRD patterns of samples were recorded at 25 °C in the 2θ angle range 5–45° with a step size of 0.04°/3 s.

## Figures and Tables

**Figure 1 marinedrugs-20-00665-f001:**
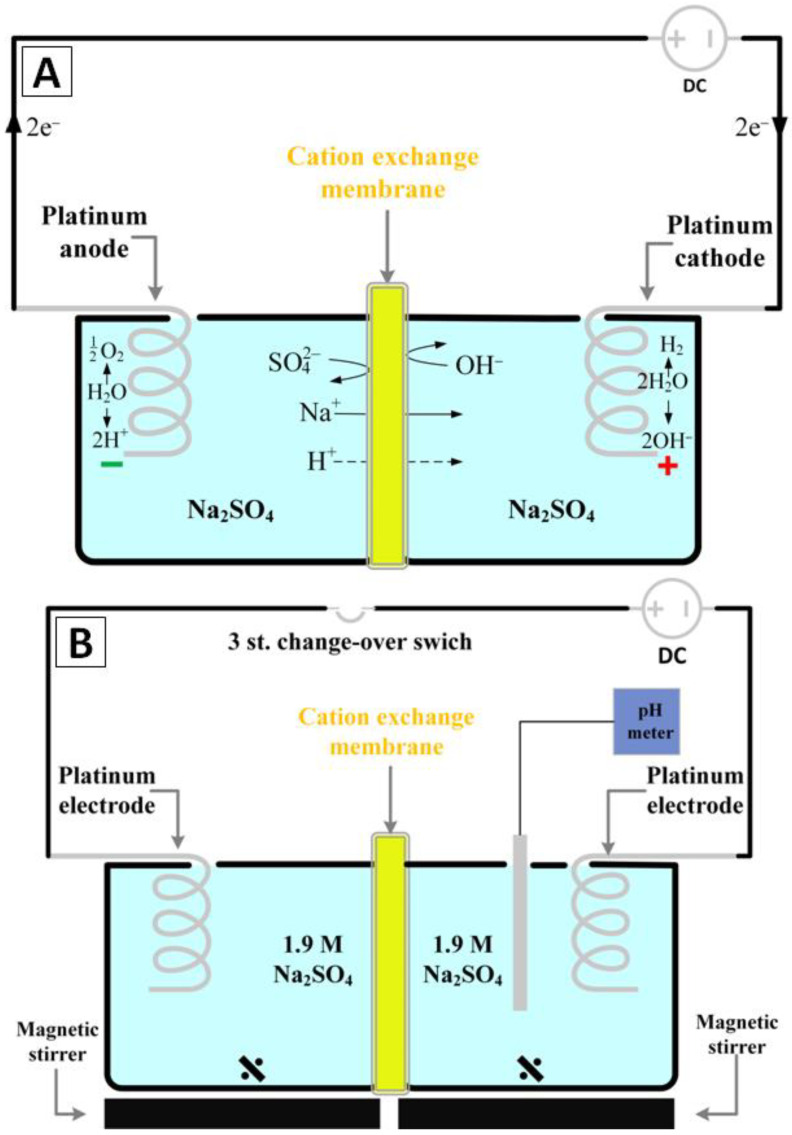
Schematic illustrations of: (**A**) the general principle of Na_2_SO_4_ aqueous solution electrolysis in the CEM electrolyzer [63] and (**B**) the experimental setup used in this study.

**Figure 2 marinedrugs-20-00665-f002:**
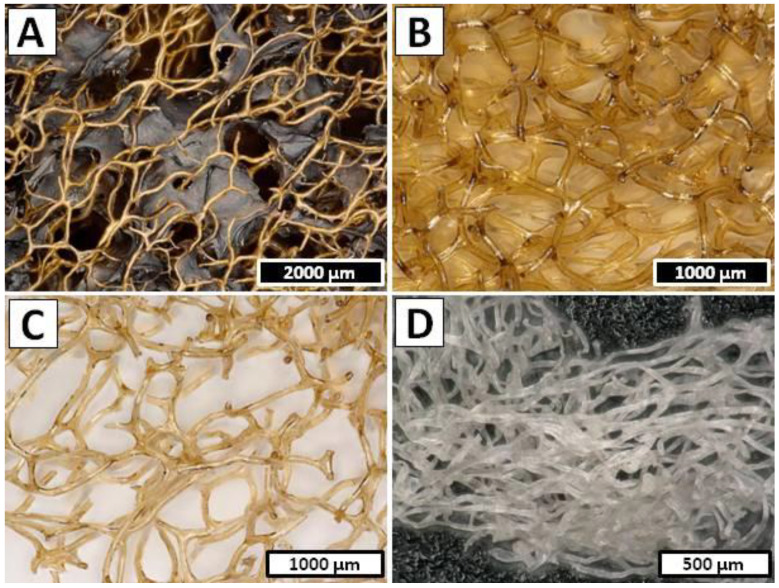
*A. archeri* sample (**A**) prior to and at different stages of electrolysis-supported isolation of chitin: (**B**) after decellularization—1.5 h of catholyte treatment, (**C**) after decalcification—1.5 h of anolyte treatment and (**D**) after deproteinization/depigmentation/desilicification step—2.5 h of catholyte treatment.

**Figure 3 marinedrugs-20-00665-f003:**
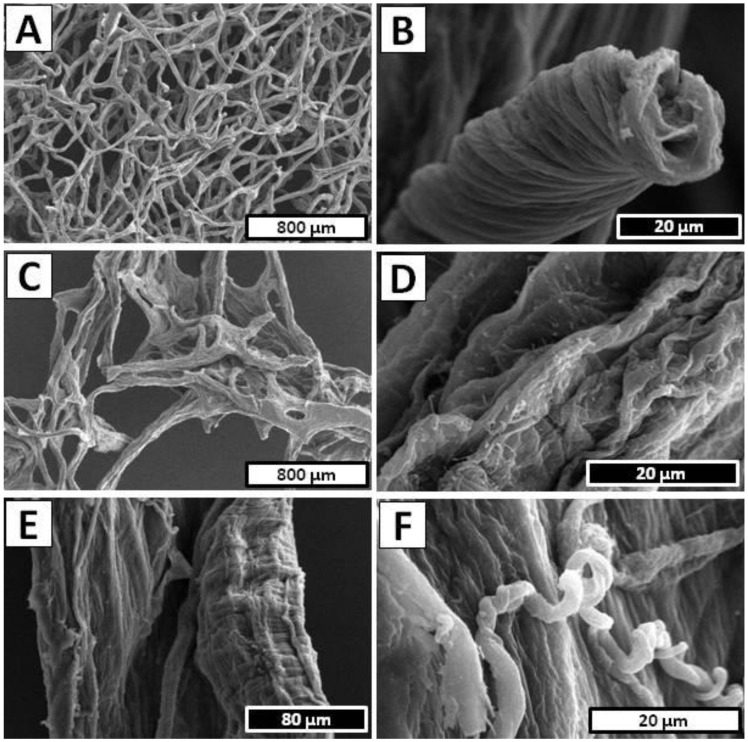
SEM imagery of the (**A**) *A. archeri*, (**C**) *I. basta* and (**E**) *S. clavata* samples after electrolytic treatment. Surface morphology of the (**B**) *A. archeri*, (**D**) *I. basta* scaffolds and (**F**) *S. clavata* fiber observed using SEM microscopy.

**Figure 4 marinedrugs-20-00665-f004:**
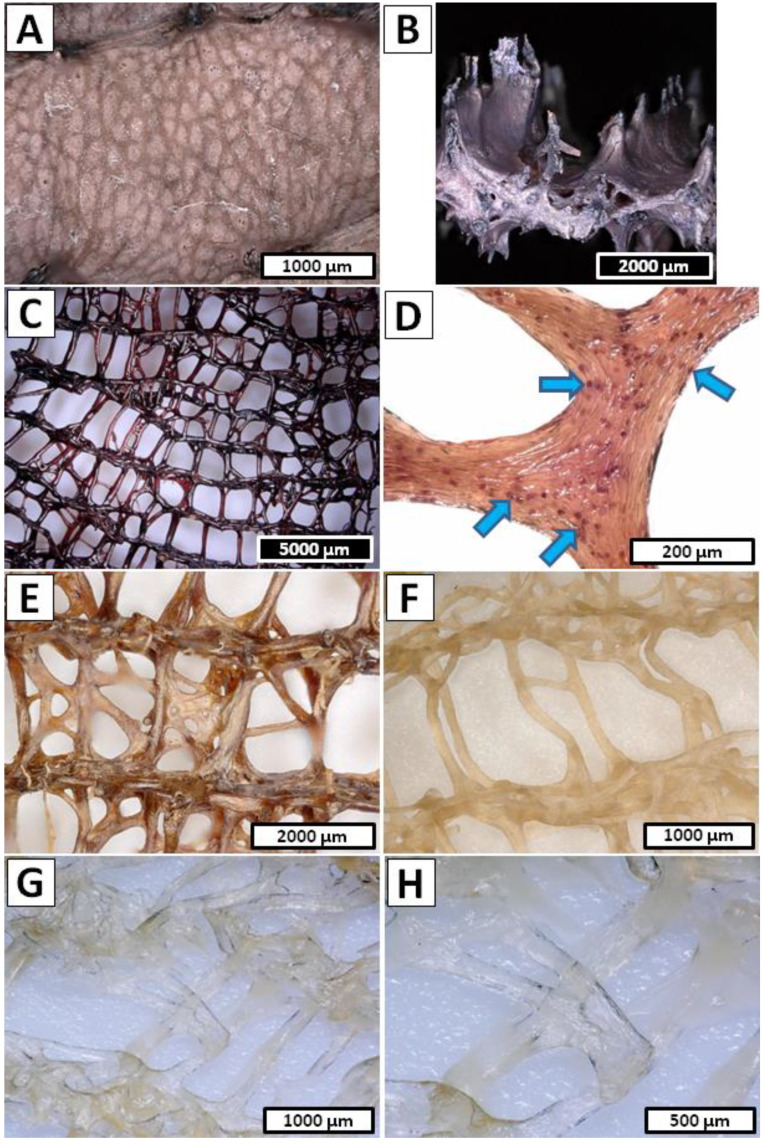
(**A**) Close-up view of the surface morphology and (**B**) cross-section of the *I. basta* sample prior to of electrolysis treatment and (**C**,**D**) after decellularization step—1.5 h of catholyte treatment. Bromotyrosines-containing spherulocytes (arrows) **(D**) can be also electrochemically removed. (**E**) *I. basta* skeleton after 1st cycle of decalcification and deproteinization/depigmentation/desilicification steps (0.5 h of anolyte and 0.5 h of catholyte treatment) and (**F**) after 8th cycles of anolyte/catholyte treatment. (**G**,**H**) Colorless chitinous scaffold isolated from *I. basta* sample.

**Figure 5 marinedrugs-20-00665-f005:**
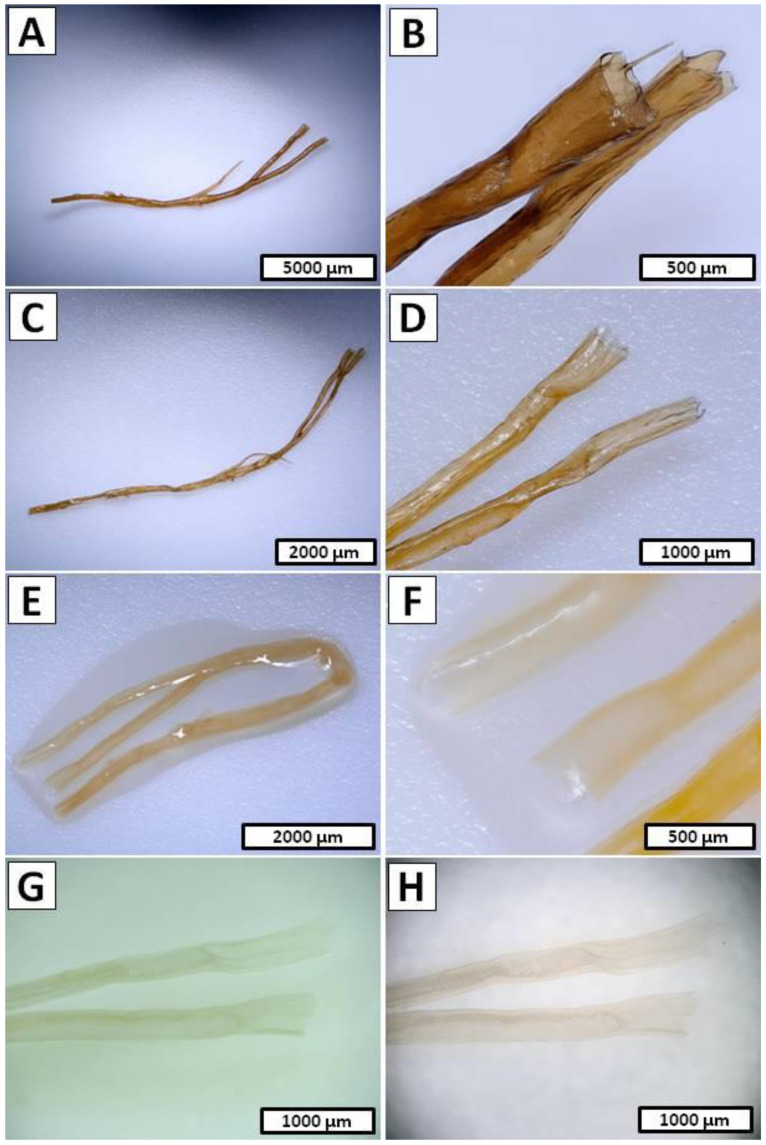
(**A**) Overview of the *S. clavata* sample and a (**B**) close-up view of the skeletal fibers before electrochemically assisted isolation of chitin process. (**C**,**D**) Sample of *S. clavata* after the 1st cycle of decalcification and deproteinization/depigmentation/desilicification steps (1 h of anolyte and 1 h of catholyte treatment). (**E**,**F**) Sample of the *S. clavata* in distilled water after 5th cycle of anolyte/catholyte treatment and (**G**,**H**) colorless chitinous tubes isolated in this study.

**Figure 6 marinedrugs-20-00665-f006:**
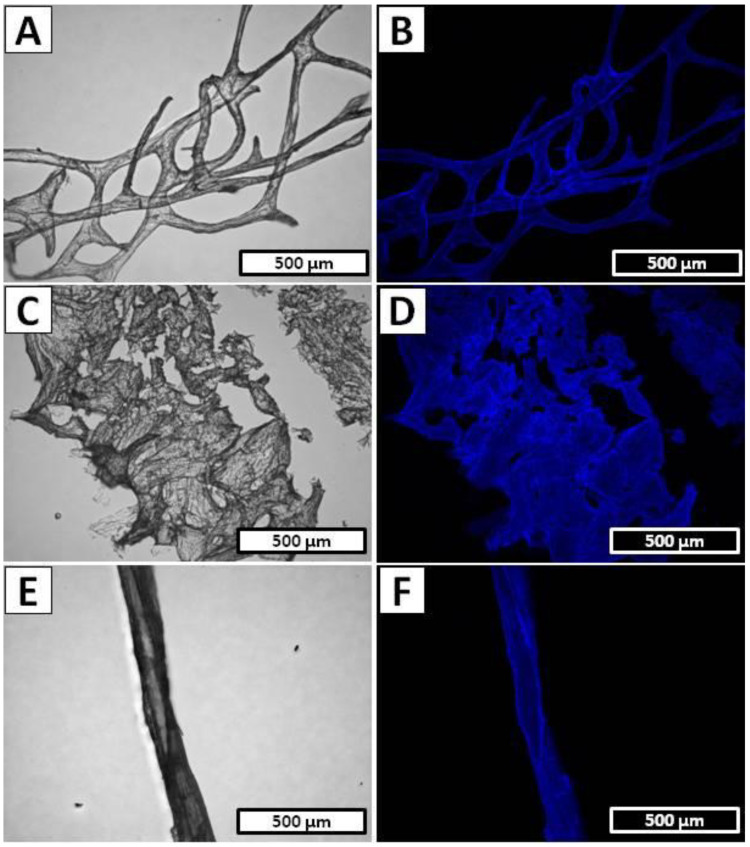
(**A**) *A. archeri*, (**C**) *I. basta* scaffolds and (**E**) *S. clavata* fiber after electrochemically-assisted isolation process. (**B**) *A. archeri*, (**D**) *I. basta* and (**F**) *S. clavata* samples after CFW staining for chitin preliminary identification (light exposure time: 1/4800 s; 1/1900 s; 1/600 s, respectively).

**Figure 7 marinedrugs-20-00665-f007:**
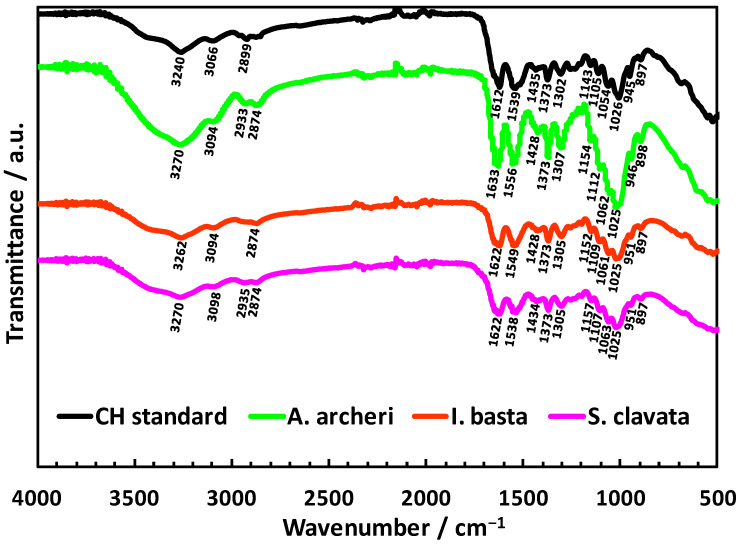
ATR-FTIR spectra of *A. archeri* (green line), *I. basta* (red line) and *S. clavata* (purple line) scaffolds after electrochemically-assisted isolation process, as well as spectra of chitinous standard (black line).

**Figure 8 marinedrugs-20-00665-f008:**
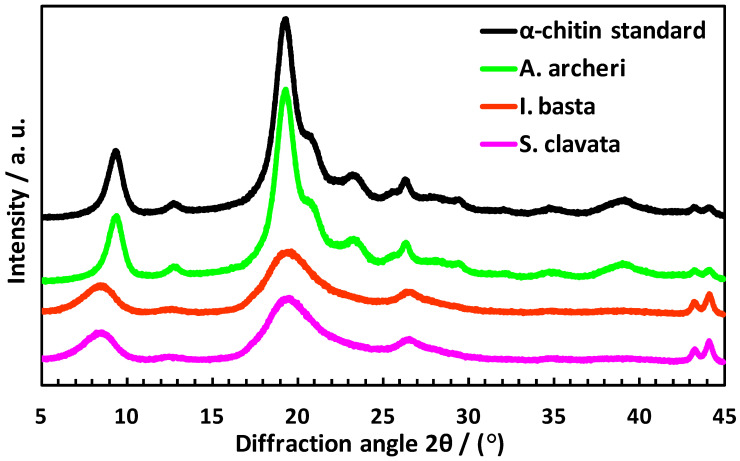
X-ray diffraction patterns of *A. archeri* (green line), *I. basta* (red line) and *S. clavata* (purple line) scaffolds after electrochemically-assisted isolation process.

**Figure 9 marinedrugs-20-00665-f009:**
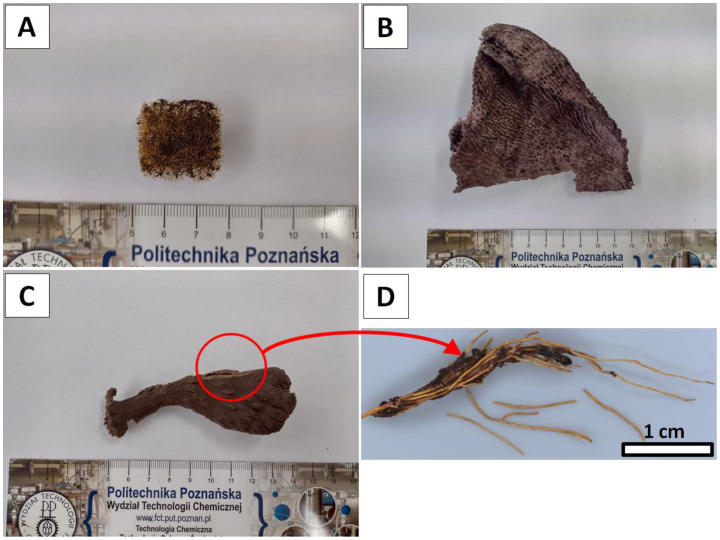
Overview of the sponge specimens used in the study. (**A**) *Aplysina archeri*, (**B**) *Ianthella basta* and (**C**) *Suberea clavata* samples. (**D**) Close-up to the skeletal fibers of *Suberea clavata* isolated from the sponge body manually.

**Figure 10 marinedrugs-20-00665-f010:**
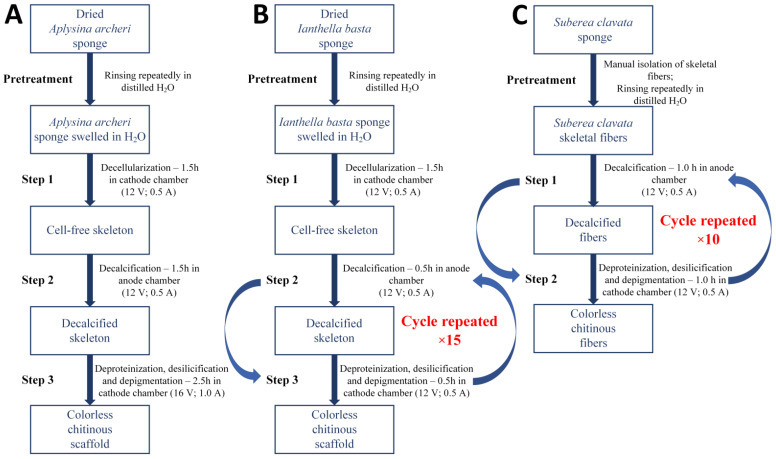
Scheme of the electrochemically-assisted isolation of chitin from demosponges (**A**) *A. archeri*, (**B**) *I. basta* and (**C**) *S. clavata*.

## Data Availability

Not applicable.

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
