# Peer review of "Electrolysis as a Universal Approach for Isolation of Diverse Chitin Scaffolds from Selected Marine Demosponges†"

_marinedrugs, 2022, doi:10.3390/md20110665_

Round 1
Reviewer 1 Report
Ref. No.: marinedrugs-1927402
Subject: Decision on Manuscript: Electrolysis as an universal approach for isolation of diverse chitin scaffolds from selected marine demosponges
Journal: Marine Drugs
Dear Editor,
I would like to thank for the invite to collaborate to review process of article “: Electrolysis as an universal approach for isolation of diverse chitin scaffolds from selected marine demosponges”. I recommend that is necessary a major revision of manuscript. Some comments are described below:
English should be improved in all manuscript.
Abstract:
“Three-dimensional chitinous scaffolds used often in regenerative medicine, tissue engi- 20 neering, biomimetics and technology are mostly isolated from marine organisms, especially marine 21 sponges (Porifera).” Although is very interesting the chitin extraction from marine sponges, the term specially should be removed, changing to such as, because this is not the most common way.
Introduction: The novelty of this research is not so clear. The authors mentioned about other authors that has already worked with electrolysis-assisted isolation of chitin, then, what is the novelty of this research? This should be very cleared in the introduction. The differences of this research and other in the literature should be mentioned.
Results:
Line 135: “The selection of such marine demosponges of Verongiida order as A. archeri, I. basta 135 and S. clavata as a chitin sources was motivated by the fact that these sponges species 136 have been relatively well described as the renewable sources of marine α-chitin in the 137 chemical extraction method as well as in the other alternative isolation approaches [6, 138 66–69].” Then, what is the novelty of this research?
Figures 5 E and F seems that some little film was formed, why? The authors should be investigated.
Figure 7 should be improved the spectra.
Discussion: More discussion is necessary. For example, the authors should be in-depthd the discussion, why the differences in extraction of chitin occurred in the three species, as the methodology were the same. In addition, other discussion, as why in the sample S. clavate a film was formed?
Materials and methods:
A section of materials in separated should be included. In addition, all reagents used, and purity should be mentioned.
In the FTIR analysis, the range, scanning, and other details should be included in the manuscript.
Similar to SEM analysis, the methodology of sample preparation, and other details should be included.
Where is the conclusion?
References: The authors should be decreased the number of references.

Author Response
Thank you for your insightful comments. All changes within the manuscript are marked in blue. For our responses please see the attachment.

Reviewer 2 Report
This study investigated the electrochemical isolation of the ready to use chitinous matrices from three species of verongiid demosponges (Aplysina archeri, Ianthella basta and Suberea clavata). The treatment times in all three cases were significantly shorter than respective in the standard chemical extraction method. Therefore, I recommend publication of this manuscript after the following minor revisions.
1. The IR spectra at entire measured wavenumbers, such as 1900-4000 cm-1 should be shown in Figure 7. The absorption peaks detected at the areas should be assigned.
2. For additional characterization of the isolated chitins, their crystalline structures should be analyzed by powder X-ray diffraction measurement.
3. According to the descriptions in Sections 2 and 3, I suggest revision of their section names to be ‘Results and Discussion’ and ‘Conclusions’, respectively.
Author Response

(The authors gave the same response as above.)

Reviewer 3 Report
The paper presents new and interesting results on the isolation of chitin from marine sources using an electrolytic method with reduction of the treatment time and conservation of the chitin structure.
The paper is scientifically valid, clearly described and the results are well presented and discussed
There are just two remarks:
1. On page 9, line 255, it is said that "all surface of the sample was covered by tangled up filaments possibly reinforcing and strengthening the mechanical properties of the main chitinous fiber". This is not supported by experimental evidence, since no mechanical tests were performed
2. Since the isolation of chitin is finalized to the preparation of scaffolds for regenerative medicine, some tests proving cytocompatibility of the isolated samples should be performed. Referring to this aspect, at least in the Conclusions, would be very useful and appreciated.
Author Response

(The authors gave the same response as above.)

Round 2
Reviewer 1 Report
Ref. No.: marinedrugs-1927402R1
Subject: Decision on Manuscript: Electrolysis as an universal approach for isolation of diverse chitin scaffolds from selected marine demosponges
Journal: Marine Drugs
Dear Editor,
I would like to thank you for the invitation to collaborate to review process of article “: Electrolysis as an universal approach for isolation of diverse chitin scaffolds from selected marine demosponges”. My recommendation is described below:
The authors did the required all corrections and the manuscript is publishable in current version.
